

# Update of the tsunami catalogue of New Caledonia using a decision table based on seismic data and maregraphic records.

Jean Roger[1], Bernard Pelletier[2], Jérôme Aucan[1]

[1]LEGOS, Institut de Recherche pour le Développement, 101, Promenade Roger Laroque, BP A5 98848 Nouméa Cedex
[2]GEOAZUR, Institut de Recherche pour le Développement, 101, Promenade Roger Laroque, BP A5 98848 Nouméa Cedex

*Correspondence to*: Jean Roger (jeanrog@hotmail.fr)

**Abstract.** 14 years ago, the December 26, 2004 Indian Ocean tsunami brought to the entire World the destruction capability of tsunamis. Since then, many research programs have been initiated to try to better understand the phenomenon and its related hazards, and to improve the early warning systems for the exposed coastal populations. Pacific Islands Countries and Territories (PICTs) are especially vulnerable to tsunamis. Amongst them, New Caledonia is a French overseas territory located in the South-western Pacific and exposed to several tsunamis sources. In 2010, a catalogue of tsunamis that were visually observed or measured in New Caledonia was published. Since this first study, several events occurred between 2009 and 2019 and an update of this catalogue was necessary within the framework of a tsunami hazard assessment project (TSUCAL). To complete this catalogue, a decision algorithm has been designed to select potential tsunamigenic events within the USGS earthquake database, using criteria on the distance to New Caledonia, the magnitude and the hypocenter depth. Then a cross-comparison between these earthquake events, the NOAA NGDC tsunami catalogue and local tide gauge records provided 25 events that were recorded in New Caledonia for the period from September 30, 2009 to January 10, 2019. These events are added to the 12 events reported with certainty during previous studies, leading to a number of 37 tsunamis triggered by earthquakes reported in New Caledonia since 1875. Six of them have been identified only thanks to local tide gauges, supporting the fact that instrumental recording of tsunamis is paramount for tsunami hazard studies, from early warning to the validation of coastal models. In addition, previously unpublished data is provided for other already reported tsunamis as well as tsunamis with unspecified date and source.

## 1 General settings

New Caledonia is located ~200 km of the Vanuatu subduction zone on the south-western Pacific Ocean between Australia and Vanuatu (Fig. 1). This subduction zone is part of the Pacific-Australia (P-A) plates boundary that runs from New-Zealand to the South to Papua-New Guinea to the North. Along this boundary the convergence rate between the Australian and Pacific plates has been estimated to 60-120 mm/yr (DeMets et al., 2010). However due to several micro-plates located between the two major plates and especially back-arc spreadings, convergence rates at trenches can largely exceeds the P-A motion and these rates were measured at up to 17 cm/yr in northern Vanuatu and 24 cm/yr in northern Tonga (Pelletier and Louat, 1989; Bevis et 1995; Pelletier et al., 1998; Calmant et al., 2003). Over the past decades, this convergence zone has clearly demonstrated its ability to generate strong shallow earthquakes, although moment magnitude does not exceed 8.2 according to the USGS catalogue.





New Caledonia is an archipelago that was originally populated by Austronesians circa 3000 years ago (Forestier, 1994). It has been discovered by Europeans on September 4, 1774 (Faivre, 1950). But according to the same author, foreigners really began to settle here only after 1840, appearing with them written reports of uncommon natural phenomena, including tsunamis. Thus, the reported written history of tsunamis covers only the last 180 years, and only the biggest events have left available information especially when associated to an earthquake felt by the population; it concerns mainly earthquakes occurring at the Vanuatu subduction zone. In addition, due to its oceanic location, New Caledonia is also exposed to tsunamis coming from other parts of the Pacific Ocean, from regional sources (e.g. Solomon and Tonga-Kermadec trenches ; Fig. 1) to transoceanic sources (e.g. Chile, Japan or Kuril subduction zones). Most of the time these tsunamis have been reported in coeval reports by witnesses or transmitted orally in the Kanak tradition and collected in a catalogue by Sahal et al. (2010) for the period running from 1875 to 2009 as an update of the previous catalogues from Soloviev and Go (1974) and Louat and Baldassari (1989).

The present study builds on the catalogue from Sahal et al. (2010) using a decision algorithm and local maregraphic data adding the events being recorded since then and adding unrevealed information concerning previously mentioned events.

## 2 Methodology

This study is based on the USGS earthquake catalogue that provides accurate information on seismic events all around the World since January 1, 1900 (U.S. Geological Survey, 2019). An algorithm has been created to select within this database the events that were potentially tsunamigenic and with potential to reach the New Caledonia coastlines. It is described hereafter.

The events extracted from the USGS database by this algorithm are then cross-compared to the reported tsunami from the NOAA NGDC tsunami database (and NOAA PTWC bulletin archives) and to local maregraphic data.

### 2.1 Search algorithm

The first step was to collect all the available earthquakes from the catalogue for the Pacific Ocean region. We decided to select only the Mw > 6.0 events according to the global tsunami databases, like the Historical Tsunami Database for the World Ocean -HTDB/WLD (http://tsun.sscc.ru/tsunami-database/index.php) or the NOAA NGDC/WDS Global Historical Tsunami Database (www.ngdc.noaa.gov/hazard/tsu_db.shtml), which empirically show that there is no tsunamis triggered by earthquakes of magnitude Mw < 6.3 (Tinti, 1991). For information, Bolt et al. (1975) indicate that the maximum run-up for a tsunami generated by a Mw = 6.5 earthquake would be no more than 0.5-0.75 m and Walker (2005) shows that tsunamigenic earthquakes with moment magnitudes Mw ≥ 8.6 had all a Pacific-wide impact. On the 10th of January 2019, this collection represents 7255 Mw > 6.0 earthquakes on a [66.205°S, 61.97°N, 118.477°W, 297.773°W] box since July 29, 1900.

Then it is important to select events able to trigger tsunami with sufficient energy to reach New Caledonia. Ward (1980) has indicated that tsunami generation is dependent upon the following criteria: the faulting mechanisms (mainly dip and slip), the magnitude (energy release) and the epicenter depth (focal depth). Thus, we consider all



the faulting mechanisms without any distinction, integrating the distance from the source to New Caledonia to the earthquake magnitude and epicenter depth in the selection algorithm.

Notice that the global tsunami catalogue is considered as a whole, and not only in the Pacific Ocean region, according to the fact that catastrophic events like the 2004 Sumatra (Indian Ocean) tsunami could be recorded by tide gauges all around the World (Rabinovich and Thomson, 2004; Titov et al., 2005).

**2.2 The faulting mechanisms**

Although thrust and normal faults are responsible of the majority of the strong subduction earthquakes and tsunamis, Tanioka and Satake (1996) have shown that in a specific case, i.e. when the rupture occurs on a steep slope with a horizontal displacement significantly larger than the vertical displacement, horizontal movement - strike-slip faulting- along a fault plane is also able to trigger a tsunami. In addition, Legg and Borrero (2001) and Borrero et al. (2004) have also shown that tectonic events occurring on strike-slip faults with sinuous traces

could trigger tsunamis by the effect of uplift and subsidence along compressional and extensional relays. Thus, we decided to consider all faulting mechanisms because they are all potentially able to exhibit a vertical component of ocean bottom disturbance, the only parameter required to disturb the water column and generate a tsunami.

**2.3 Relationship between earthquake magnitude and focal depth and tsunami generation**

We use the data from NOAA NGDC/WDS Global Historical Tsunami Database to plot the earthquake focal depth as a function of magnitude for 744 tsunamigenic earthquakes around the World from July 21, 365 to August 24, 2018 (Fig. 2). Only 47 (6.3 %) of these events have Mw < 6.0 and 32 (4.3 %) have been located at a depth of more than 100 km. Considering that these events are not located within the Pacific Ocean and/or they did not produce a sufficient tsunami to be recorded in New Caledonia, we decided to look only at earthquakes of

magnitude Mw > 6.0 and focal depth < 100 km in the following. Note that these values are consistent with different Early warning systems criteria (UNESCO/IOC, 2009).

**2.4 Tsunami amplitude and distance from the source**

The tsunami amplitude, its wavelength and frequency components, directly linked to the faulting mechanisms, determine the extent of the impacted zone: local, regional or ocean-wide impact. In fact, a tsunami triggered by a

landslide or a moderate earthquake is more inclined to dispersive phenomenon than a tsunami triggered by a large earthquake because dispersion is directly linked to the wavelength and frequency content, the water depth and the propagation distance (Glimsdal et al., 2013). In fact, smaller the source is, quicker is the energy decay and finally the disappearance of the tsunami waves over the time and distance from the source (Rabinovich et al., 2013). For example, Tanioka et al. (2018) show the role of this dispersion phenomenon when looking at the

DART buoy record of the tsunami triggered by the Mw 6.9 2016 Salvador-Nicaragua earthquake: in that specific case, the linear long waves theory overestimates considerably the numerical modeling results in comparison to the results obtained by the help of linear Boussinesq equations, taking into account dispersion effect.

Considering that the local and regional sources (from the Solomon, Vanuatu and Tonga-Kermadec subduction zones) are located within a circle of radius ~2500 km (Fig. 3) and that the far-field potential tsunami sources are

located between 4400 km (for the Mariana Trench, offshore Guam) and more than 10000 km away from New





Caledonia (for Chile subduction zone), we decided to make an event sorting upon this distance criterion of D = 2500 km.

**2.5 Construction of the algorithm**

The algorithm is based on the previous explanations and lays down the rules to keep or reject a considered event
from the USGS catalogue with respect to specific conditions on three different parameters: the earthquake magnitude (M), the focal depth (F) and the distance between the source and New Caledonia (D). Four cases are considered to select whether an event is kept or rejected. They are summarized on figure 4.

For the calculations of distance from an earthquake epicenter to New Caledonia on a sphere, an approximate theoretical center of New Caledonia has been determined calculating the barycentre of a triangle made with the
three following points: [163.576030, -19.539454°] for the northernmost point of the archipelago, [167.570644, -22.762149] for the southernmost point and [168.135799°, -21.450552°] for the easternmost point. Its coordinates are [166.427491, -21.250718].

**2.6 Sea-level data**

The tide gauge stations of New Caledonia are located on figure 5. All the tide gauges were installed along the
East Coast of the Grande Terre of New Caledonia and in the Loyalty Islands, except the historical tide gauge in Nouméa (Chaleix then Numbo), the territory capital located on the West coast of Grande Terre. (Aucan et al., 2017). Because of its location on the opposite side of all the reported main incoming tsunami pathways - azimuthal direction window from ~304°N (Papua New Guinea) to~160°N (New Zealand)-, it is likely impossible to record small tsunamis (generally arbitrarily < 1 m) on this tide gauge. Instrumental records of hourly sea level
for Nouméa has been back extended to 1957 (Aucan et al., 2017), and in the present paper we show previously unpublished records of digitized high frequency sea level data.

All the tide gauges started recording high frequency sea level (with sampling rates faster than 5 minutes) only after 2010 or later. The tide gauge characteristics are summarized on table 1.

In addition to these tide gauges, several pressure gauges have also been installed by the IRD in Poindimié,
Ouvéa and Uitoe (Fig. 2) within the framework of the ReefTEMPS project (Varillon et al., 2018) or the EMIL project (Aucan et al., 2018). Their characteristics are also summarized in table 1.

**3 Tsunami catalogue**

**3.1 Results of extraction**

The algorithm has been able to extract 1337 events from the USGS earthquakes catalogue between July 29, 1900
and January 10, 2019. If the magnitude threshold is upgraded from 6.0 to 6.3 (Mw ≥ 6.3) the number of extracted events is reduced to 967. Thus according to the low potential of such earthquakes to trigger tsunamis, only earthquakes of magnitude Mw ≥ 6.3 are considered afterwards.

At first we decided to look only at the events that occurred after the period considered by Sahal et al. (2010), i.e. after the September 29, 2009 Samoa tsunami. It represents 113 events from the previous extraction (113/967).
But not all these events triggered a tsunami and even less a tsunami able to reach New Caledonia. To look at their tsunamigenic capabilities, this list of earthquakes was cross-compared with the NOAA NGDC/WDS Global



Historical Tsunami Database which provided 44 events that were reported as tsunamigenic (last download date : August 24, 2018). This list includes events like the February 27, 2010 Chile Mw 8.8 or the March 11, 2011 Japan Mw 9.1 earthquakes. The 44 tsunamigenic events represents 38.94% of the 113 events extracted form the

USGS EQ database. From August 24, 2018 to January 10, 2019, 9 events have been extracted from the USGS database with the algorithm and only 2 of them where followed by a NOAA PTWC bulletin indicating they triggered a tsunami.

Thus, there were 46 (44 + 2) tsunamigenic events for the period from September 29, 2009 to January 17, 2019, corresponding to 40.71% of the 113 identified earthquakes. From these 46 events, only 14 have been followed

by a tsunami recorded in New Caledonia according to the observations included within PTWC bulletins.

An in-depth look at the other 32 events (46 - 14) case by case using both Tsunami Database and newspapers coupled to tide gauges data from the 8 local stations has been performed to identify small tsunamis that could have been missed by PTWC analysis (i.e., not mentioned in bulletins).

From September 2009 to February 2011, 8 extracted tsunamigenic earthquakes have been reported by PTWC

bulletins. As mentioned above, on this period only the Nouméa tide gauge operated at high enough frequency. But the analysis of the recorded signals at the Nouméa tide gauge for those 8 events, especially the Mw 8.8 Bio-Bio, Chile, earthquake of February 27, 2010, recorded in nearby islands (Tonga, New Zealand, Australia, French Polynesia, etc.), did not reveal any tsunami.

From February 2011 to January 2019, there are still 24 (32 - 8) tsunamigenic events extracted from the list.

Hienghène and Ouinné tide gauges have been chosen to identify the corresponding recorded signals since these two tide gauges located on the East coast of the Grande Terre (Fig. 5) are exposed to several tsunami sources, and are located in bays or estuaries that amplify tsunami signals. 7 of these 24 tsunamis have been recorded by Ouinné gauge but not reported in New Caledonia by PTWC. Among these 7 events, 2 did not trigger a  PTWC bulletin (October 31, 2017 and November 01, 2017 events) although they are reported within the NOAA NGDC

catalogue.

The same analysis of local tide gauge signals for the 14 events reported by PTWC in New Caledonia showed that two of them were not recorded locally with certainty (probably an ambiguity with the background noise): the July 18, 2015 Solomon and August 20, 2011 Vanuatu events.

Thus, there are 19 (12 + 7) tsunamis having been recorded in New Caledonia from September 30, 2009 to

January 10, 2019 with certainty. These 19 events are detailed on table 2.

An in-depth analysis of the sea-level data recorded at the Ouinné and Hienghène tide gauges for events extracted by the algorithm but not reported as tsunamigenic in the NOAA NGDC catalogue allows to find additional tsunamis that were recorded in New Caledonia between September 30, 2009 and January 10, 2019. Thus, on the 67 events (113 - 46) not identified as tsunamigenic by NOAA NGDC, 6 have still produced tsunamis recorded

on New Caledonia gauges. They are reported on table 2.

These 25 events represent a considerable update of the catalogue from Sahal et al. (2010). It is important to note that aftershocks of powerful tsunamigenic earthquakes as the Mw 7.8 Kirakira, Solomon, earthquake of December 8, 2016, could also trigger tsunamis that would be drowned within the main shock tsunami signal. Also small tsunami signals could be covered by the background noise, especially the coastal or offshore

infragravity waves (Stephenson and Rabinovich, 2009, Aucan and Ardhuin, 2013).





For the period before September 29, 2009, Sahal et al. (2010) have collected 18 events, including 12 events related with certainty to an identified earthquake. These 12 events are detailed on table 3. There is a strong uncertainty on the accuracy of the 6 other reported events concerning the date as well as the source (see section c) below).

11 out of the 12 seismic events reported by Sahal et al. (2010) and shown in table 3 have been kept with the algorithm; the first one from 1875 is out of range from the USGS available database beginning on January 01, 1900. These 11 events are part of the 854 extracted events by the algorithm for the period from January 01, 1900 to September 29, 2009 (included). It represents less than 1.29% of earthquake of magnitude $Mw \geq 6.3$ being able to produce a tsunami reaching New Caledonia over this period.

In comparison, for the period from September 30, 2009 to January 10, 2019, 25 of the 113 extracted events with the algorithm have produced a tsunami having reached New Caledonia. This represents 22.12% of the earthquakes of magnitude $Mw \geq 6.3$.

Even if we consider only the tsunamis reported in the NOAA NGDC catalogue, it corresponds respectively to 11 out of 854 events for the period from January 01, 1900 to September 29, 2009 (included) and 14 out of 113

events for the period from September 30, 2009 to January 10, 2019, i.e. 1.29% and 12.39%. Nearly an increase by a factor of ten.

To compare exactly the 2 periods, we should consider the same number of days, i.e. 9 years, 3 months and 10 days. Thus from June 19, 2000 to September 29, 2009 the algorithm extracted 111 events potentially able to trigger tsunamis able to reach New Caledonia (this number is very close to the 113 events extracted after

September 29, 2009 and to the 96 events extracted from march 9, 1991 to June 19, 2000). Cross-comparing those 111 events to the NOAA-NGDC tsunami database, it appears that 26 earthquakes triggered a tsunami on the Pacific Ocean, either local, regional or transoceanic. And from these 26 events, only 4 have been reported in New Caledonia according to Sahal et al. (2010).

### 3.2 Individual events during the 2009-2019 period

During this period, 4 events are particularly interesting as their records demonstrate the importance of  local tide gauges and pressure sensors on tsunami hazard studies.

### 3.2.1 February 6, 2013 Solomon tsunami

The tsunami generated by the Santa Cruz, Solomon, Mw 8.0 earthquake of February 6, 2013 at 01:12:25 UTC was recorded at the Lifou tide gauge near 3 a.m. UTC (2 p.m. local time)  and well observed by local witnesses

(Fig. 6). The two pictures shown on figure 6 have been taken during the first wave maximum (a) and at the following minimum (b).

### 3.2.2 September 16, 2015 Chile tsunami

On September 16, 2015 at 22:54:32 UTC a magnitude Mw 8.3 earthquake in the region off Illapel, Chile triggered a transoceanic tsunami. After hours of propagation, it reached most of the South Pacific Ocean tide

gauges. In New Caledonia it has been recorded on permanent tide gauge stations and pressure gauges about 16 hours after the earthquake as shown on Figure 7. This event is particularly interesting in the sense it confirms clearly that tsunamis are amplified near the Ouinné tide gauge, more so than at the other gauges (except



Hienghène where unfortunately the beginning of the record is missing). At Poindimié two pressure gauges installed outside of the lagoon (Poindimié_Fourmi) and inside the lagoon close to the shore (Poindimié_Tieti)

also recorded the tsunami. Data from the two gauges shows the wave shoaling  probably also the amplification (up to 5 times) due to resonance inside the lagoon. Also, during this event a pressure gauge was rapidly installed in the Chaleix Naval Base at the location of the discontinued Chaleix tide gauge to compare the recorded signal to the Numbo tide gauge signal. This emergency deployment allows to highlight the fact that there is a clear site effect in Nouméa's Bay, the signal at Chaleix being nearly three times more important than at Numbo.

### 3.2.3 November 19, 2017 South Vanuatu tsunami

Another example is given by three earthquakes that occurred during the November 19, 2017 seismic crisis located East of Mare Island: two very small tsunamis have been triggered by the Mw 6.3 and 6.6 ,foreshocks of the Mw 7.0 earthquake, which triggered a tsunami reaching a maximum amplitude of 0.8 m at Ouinné tide gauge. Despite their small amplitude, they are very well recorded and shown on figure 8.

### 3.2.4 December 5, 2018 Vanuatu tsunami

On December 5, 2018, a magnitude Mw 7.5 earthquake occurred at 04:18:08 UTC in the South of the Vanuatu subduction zone. Widely felt by the population in New Caledonia but also in the Vanuatu, a tsunami was soon recorded, firstly on the Loyalty Is. tide gauges (Maré and Lifou) and on all the other tide gauges within 1h30 after the main shock (Fig. 9). As it reached sometimes more than 1 m high according to eye-witnesses, this

tsunami was also observed by numerous people for example in Yaté, close to Ouinné, on the Southeast coast of Grande Terre (Fig. 10a), and on the East coast of the Isle of Pines around the Méridien Resort and the Natural Pool touristic site where people have been evacuated (Fig. 10b).

### 3.3 Additional information on previously reported events

Five other events could be likely added to the herein catalogue. They come from testimonies and regional

records (some of them have been already discussed in Sahal et al. (2010)).

### 3.3.1 Testimonies of tsunami events with unspecified date and not closely linked in time with any earthquake

The «1936» event reported in Northeast of the Grande Terre (north of Hienghène) with a 2 to 3 m runup could be due to a local landslide or possibly may be linked to the July 1934 North Vanuatu earthquake (Mw 7.8) which

has triggered a tsunami observed in the same region (Hienghène-Touho).
The «May-July 1942 or 1943» large wave reported in Hienghène (2.5 m runup) and possibly the flooding «around 1940» reported on Isle of Pines (2 m runup) could be attributed to the same major event (although link between these reports is uncertain). No link could be made between these time periods and any earthquake. These events can be the result of landslides.

### 3.3.2 Testimonies of events with unspecified date but that can be linked to major local earthquake

The «December 1950 – February 1951» swelling and tidal wave reported at different localities along the east coast of the Grande Terre (Hienghène, Poindimié, Ponérihouen, Canala) and on Isle of Pines could be linked to





the December 2, 1951 South Vanuatu large earthquake (Mw 7.9) which is one the largest events located close to New Caledonia and having generated a tsunami observed in Port Vila, Vanuatu.

**3.3.3 No record or testimony of worldwide or basin-wide events which are recorded especially in the vicinity of New Caledonia**

The December 26, 2004 Indonesian tsunami (Mw 9.1 Sumatra earthquake) and the February 27, 2010 Chilean tsunami (Mw 8.8 Maule earthquake) were not reported in New Caledonia : there was only one tide gauge operating in Nouméa in 2004 (e.g. Chaleix, sampled at a frequency of 1 hour) and in 2010 (e.g. Numbo, sampled 265 at a frequency of 10 min)) although they have been well recorded in Tonga, New Zealand and Australia.

**3.3.4 May 22, 1960 Chile tsunami**

Although it has been already included within the catalogue from Sahal et al. (2010), the Great 1960 Chile tsunami was only reported through witness observations. Here we present a historical maregraphic record recovered in the SHOM archive, that shows this transoceanic tsunami recorded in Nouméa by the Chaleix tide 270 gauge (Fig. 11). The tsunami amplitude is about 30 cm for the two primary waves.

**4 Discussion and conclusion**

**4.1 Limitations of the methodology**

This methodology using an extraction algorithm is based only on the tsunamis generated by earthquakes and thus, do not considered tsunami triggered by landslides or, less frequently, by volcanic eruptions, representing 275 respectively about 7% and 5% of the reported tsunamis according to Harbitz et al. (2014). In the available data, there is no evidence of tsunamis related to active volcanism or landslides. Anyway, active submarine volcanoes exist in the neighborhood of New Caledonia, especially off the Loyalty Is. (Gemini seamounts, 200 km East of Maré and South of Aneytum, Vanuatu) on the Vanuatu volcanic arc. Besides, numerous submarine landslides have been mapped along the margins of Grande Terre and Loyalty Is.

Another limitation comes from the fact that we do not consider the tsunami amplitudes in this study. In fact, it is very difficult to estimate a maximum value for each event in New Caledonia, because of obvious lack in field observations for each one and absence of tide gauge in specific places, like the Isle of Pins where important tsunami run-ups have been reported for at least the December 5, 2018 event. A tsunami could have been lowly recorded by a tide gauge located in a protected area, for example in Nouméa harbor, although it has a strong 285 impact on an exposed coast, for example on the East coast of the Isle of Pines. In addition, as detailed by Ioualalen et al. (2017), resonance phenomenon seem to play a predominant role on the tsunami behavior, especially in the Loyalty Is., depending directly on the source location and geometry. So, information concerning maximum observed amplitudes mentioned in table 2 and 3 gives just an idea of what happened during the reported events.

**4.2 Contributions for tsunami hazard assessment and risk management**

The 25 events are added to the 12 events from Sahal et al. (2010), leading to a list of 37 tsunamis reported/recorded for New Caledonia over the last 144 years. Besides the 1875 event (no exact location





available), the 36 earthquake epicenters are shown on figure 12. As expected, it highlights 5 different
tsunamigenic zones able to trigger tsunamis toward New Caledonia: locally, the Vanuatu subduction zone is
responsible of 17/37 tsunamis, i.e. 45.94 %; at a regional scale, the Tonga-Kermadec subduction zone triggered
3/37 tsunamis, i.e. 8.1 % and the Solomon / Papua New Guinea subduction zone is responsible of 9/37, i.e. 24,32
%; and at an ocean scale the transoceanic tsunamis represents 8/37 events, i.e. 21.62 %.

Besides the December 17, 2016 PNG Mw 7.9 earthquake which hypocenter has been located ~100 km deep, the
other earthquakes are located not deeper than 50 km.

Concerning tsunami amplitudes, their observed range varies to a few centimeters to several meters. Local
tsunamis issued from South Vanuatu earthquakes impact mainly the Loyalty Is. and the South-east coast of
Grande Terre (including the Isle of Pines) and are the most frequent and stronger. The north-eastern part of
Grande Terre is more impacted by regional tsunamis coming from the North (Solomon and North Vanuatu
subduction zones). Transoceanic tsunamis are also important to be considered in New Caledonia, able to produce
locally wave amplification of 1-2 m high.

A graphic representation of the 36 earthquakes showing magnitude (Mw) as a function of the distance between
the epicenter location and the center of New Caledonia highlights 3 different groups (Fig. 13): the group of 8
earthquakes on the right corresponds without surprise to the transoceanic tsunamis; the group on the left
corresponds to the local events from the southern Vanuatu subduction zone and the central group corresponds to
the regional events (Solomon, Northern Vanuatu and Tonga). In terms of risk managing, this study brings new
constraints for alert thresholds:

- In a local field, with an epicenter located within a distance less than 500 km, only a magnitude Mw ≥ 6.3
earthquake is able to trigger a tsunami toward New Caledonia coastlines.

- At a regional scale, i.e. at a distance of more than 1000 km, only the earthquakes with magnitude Mw ≥ 6.7
would be considered as potentially hazardous for New Caledonia in terms of tsunami waves and currents.

- At a far field location, i.e. at a distance of more than 6000 km, only earthquakes with magnitude Mw ≥ 7.7
would be considered as potentially hazardous for New Caledonia in terms of tsunami waves and currents.

### 4.3 Conclusion

This study allows to complete the tsunami catalogue of New Caledonia with 25 new events of seismic origin for
the period between September 30, 2009 and January 10, 2019: 19 already identified in the NOAA NGDC
tsunami catalogue and 6 others recorded on New Caledonia gauges but not reported either in this catalogue not
within the NOAA PTWC bulletins. The catalogue is now reaching a number of 37 tsunamis.

It also emphasizes that there is a considerable lack of tsunami information in New Caledonia for the pre-
September 2009 period concerning medium-magnitude events, due to the fact that there was less or no tide
gauges and DART buoys able to record small tsunamis and also because there were less people living close to
the shores.

In addition, this study highlights clearly the value of tide gauges records, including old paper ones, and the
necessity to settle the gauges in well identified locations, i.e. not always in sheltered areas but more in places
facing main tsunami pathways. It also brings to light the necessity to add more sensors in exposed areas like on
the East coast of the Isle of Pines.



**Acknowledgements**

This study is part of a national project of tsunami hazard assessment funded by the local government of New Caledonia. The authors would like to thank the SHOM for providing tide gauge data (http://doi.org/10.17183/REFMAR), and the ReefTEMPS coastal sea waters observation network for providing
pressure gauge data (http://doi.org/10.17882/55128).

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



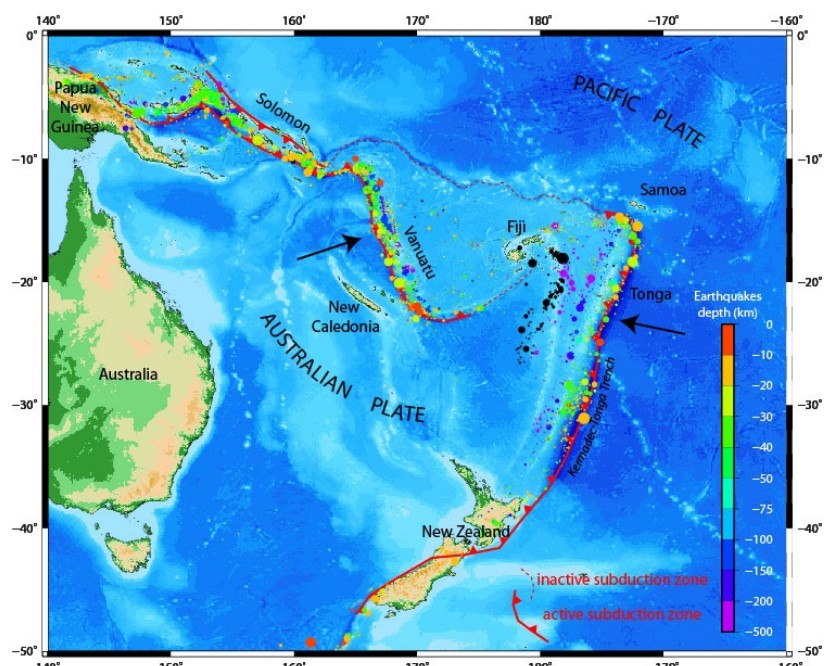

**Figure 1: Regional tectonic settings around New Caledonia. The colored dots represent Mw ≥ 6.0 earthquakes recorded since January 01, 1900. The black arrows symbolize the relative motion of the Australian and Pacific plates.**

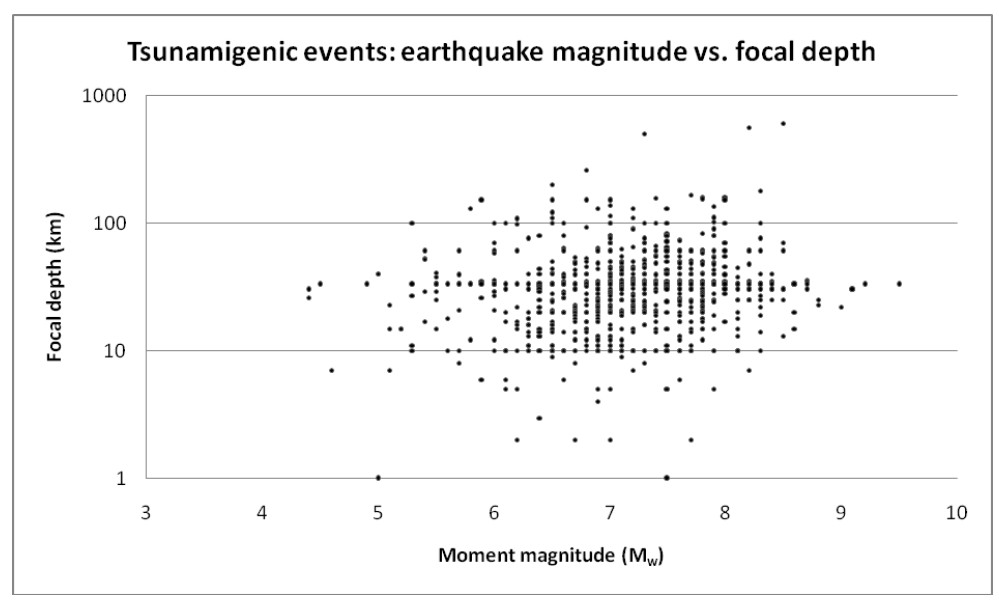

**Figure 2 : Relationship between earthquake focal depth and moment magnitude for 744 tsunamigenic events of the**
**NGDC/WDS Global Historical Tsunami Database.**



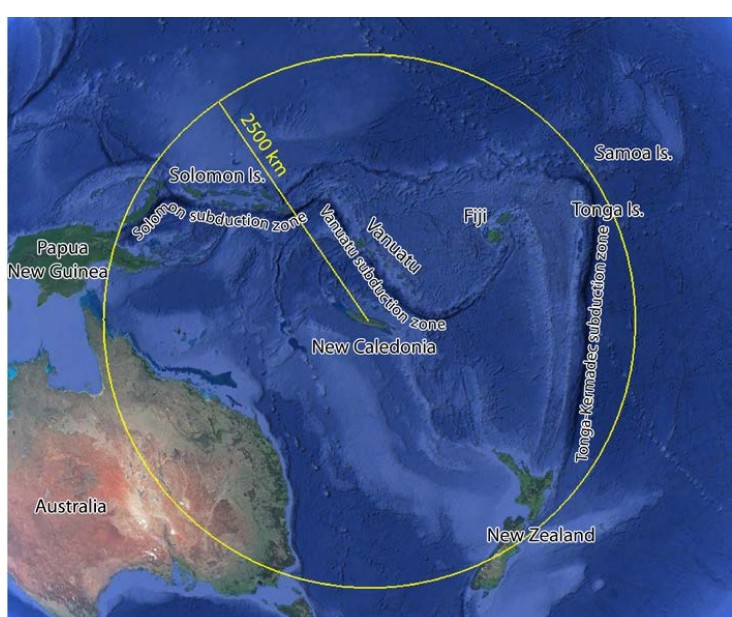

**Figure 3 : 2500 km around New Caledonia (Credit: Google 2018, Landsat/Copernicus Image).**

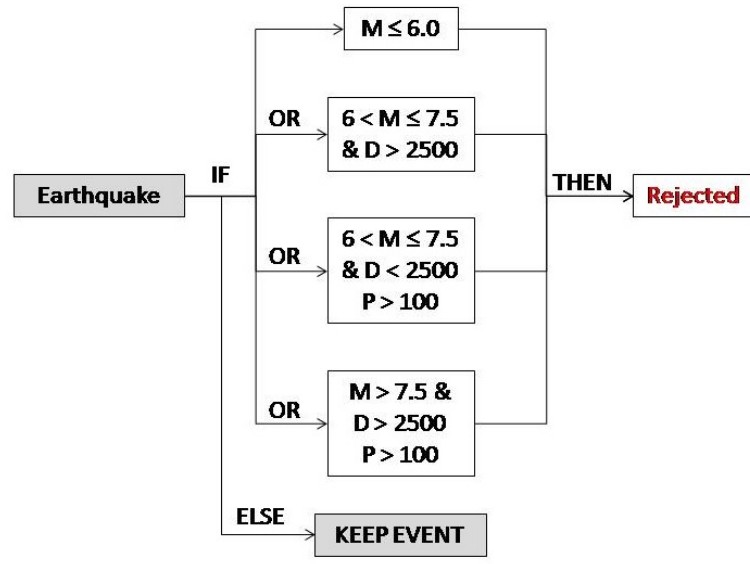


**Figure 4 : Decision algorithm to select events automatically.**



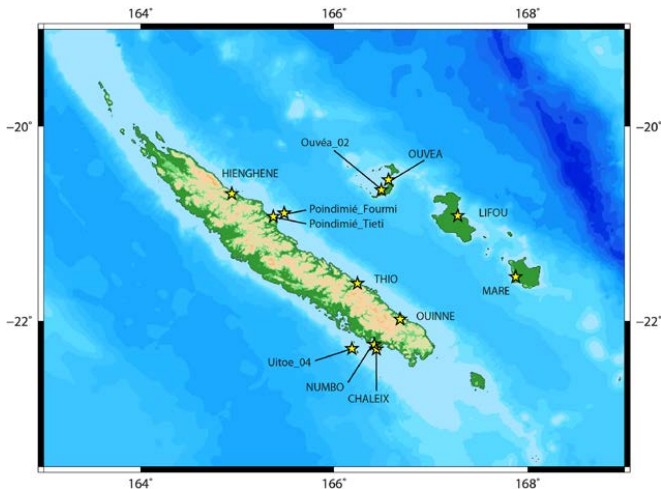

**Figure 5: New Caledonia tide and pressure gages.**


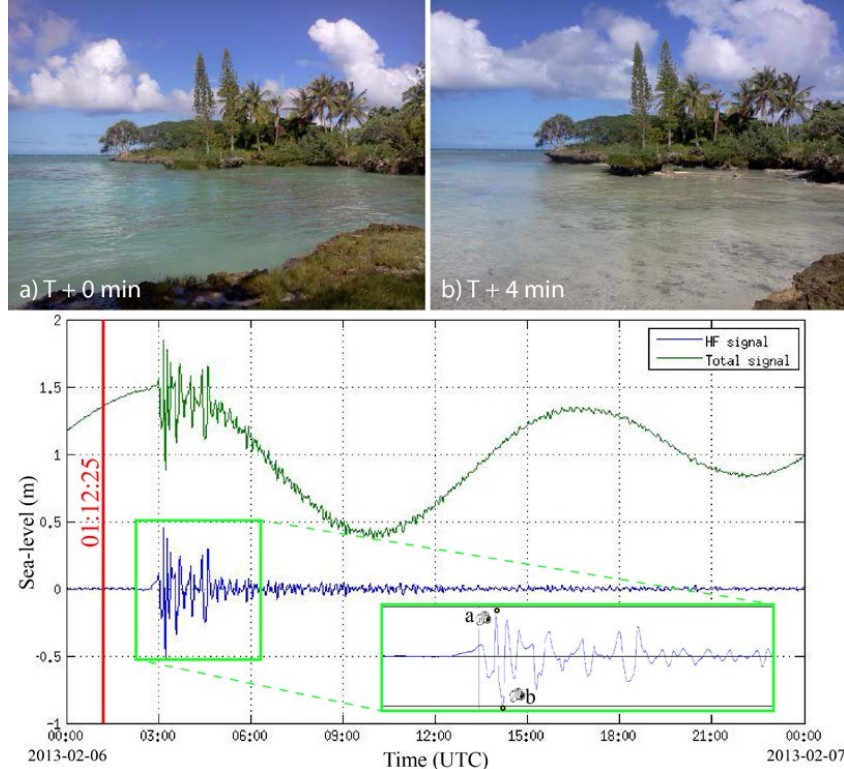

**Figure 6 : The February 6, 2013 Solomon tsunami record on Lifou tide gauge (green: raw signal; blue: filtered). The red line locates the earthquake time. The two pictures have been taken at Mou (on the South-eastern coast of Lifou) at the maximum (a) and minimum (b) sea levels and located on the green inset focusing on the main tsunami signal**

**(Photos courtesy of Matthieu Le Duff).**





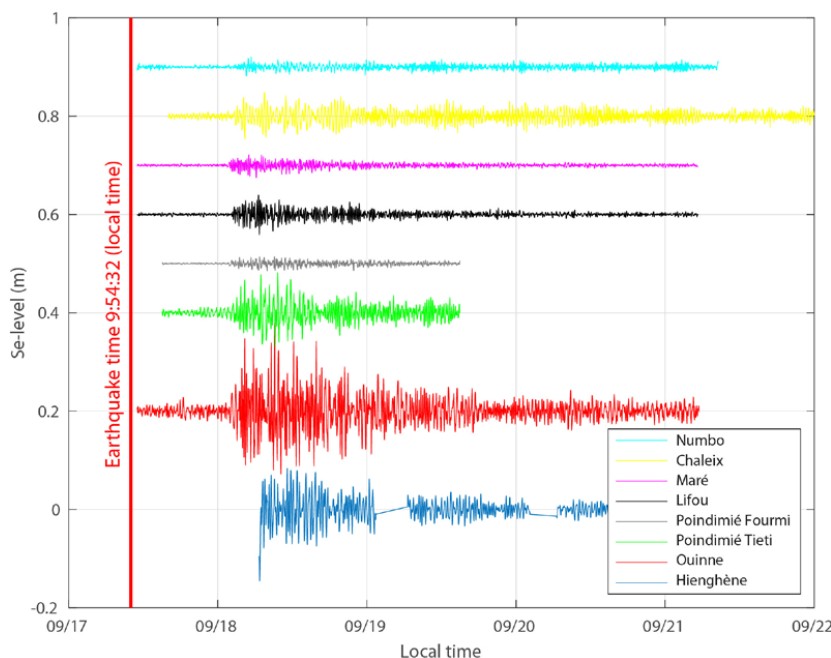

**Figure 7 : Sea level variations recorded on tide and pressure gauges in New Caledonia following the 2015 Illapel, Chile, earthquake.**

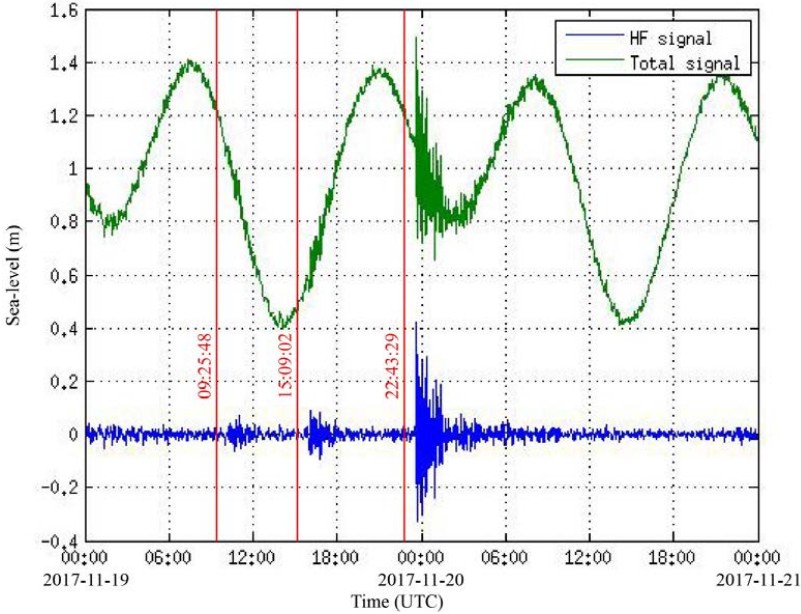

**Figure 8 : The triplet of tsunamis of November 19, 2017 recorded at Ouinné tide gauge. On the blue graphic the tide signal (visible on the green graphic) has been filtered. The red lines locate the times of the three different earthquakes.**




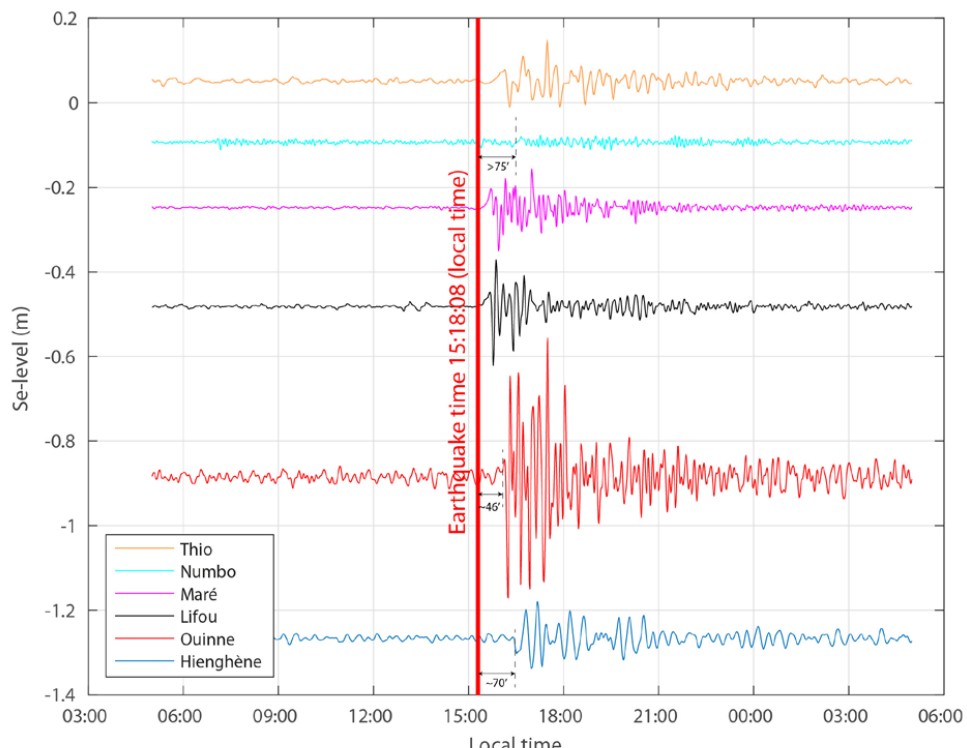

**Figure 9 : Sea level variations recorded on tide and pressure gauges in New Caledonia following the 2018 Vanuatu earthquake.**

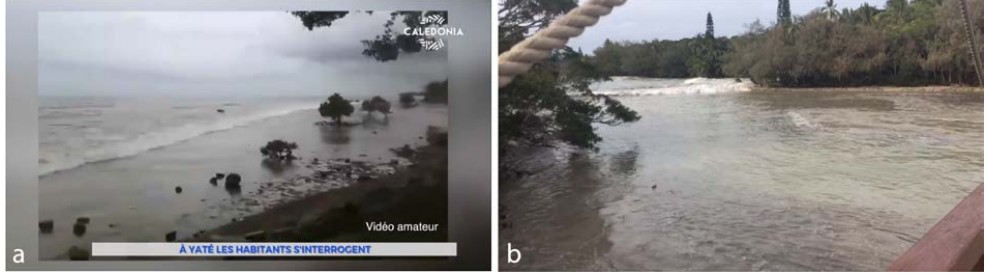


**Figure 10 : Arrival of the December 5, 2018 tsunami in a) Yaté after the first withdrawal of the sea (Courtesy of Rose-Mai Néa) and b) at the bridge linking the Méridien Resort island to the Isle of Pines (Courtesy of Moana Bretault).**





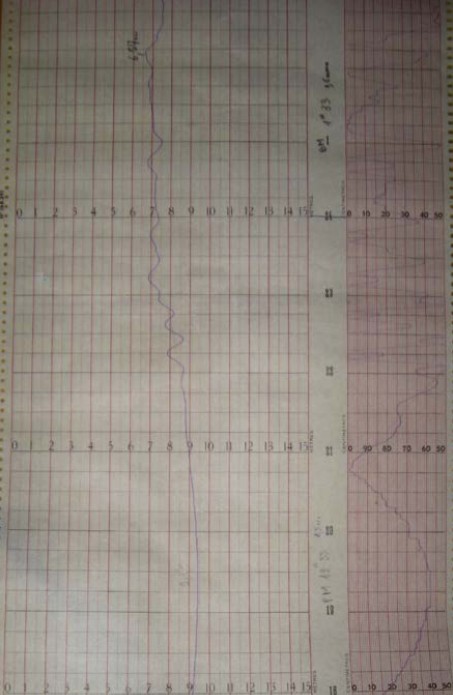

**Figure 11 : Tide gauge record of the 1960 Chile tsunami in Nouméa (Chaleix station): original record (left) and digitized signal (right), total (in green) and detided (in blue).**

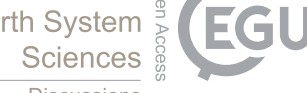

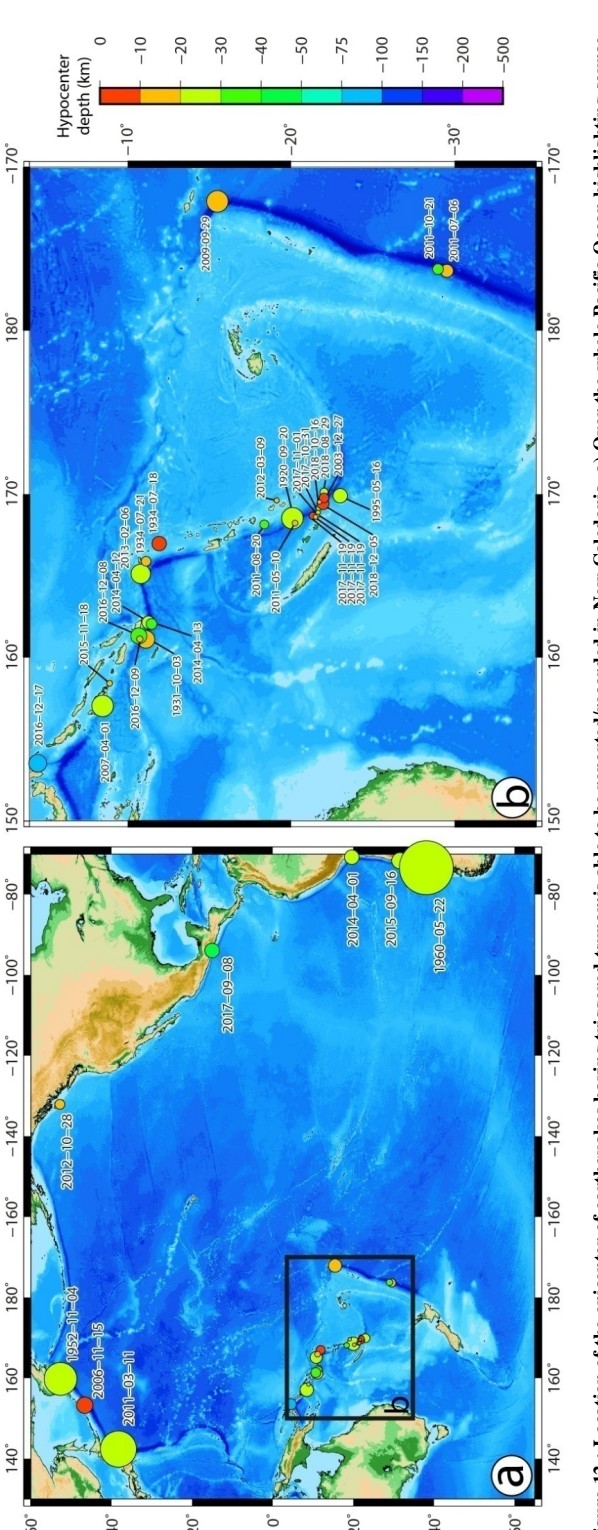

**Figure 12 :** Location of the epicenter of earthquakes having triggered tsunamis able to be reported/recorded in New Caledonia. a) Over the whole Pacific Ocean highlighting source locations of transoceanic tsunamis ; b) at a regional scale, highlighting regional and local sources.

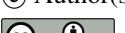



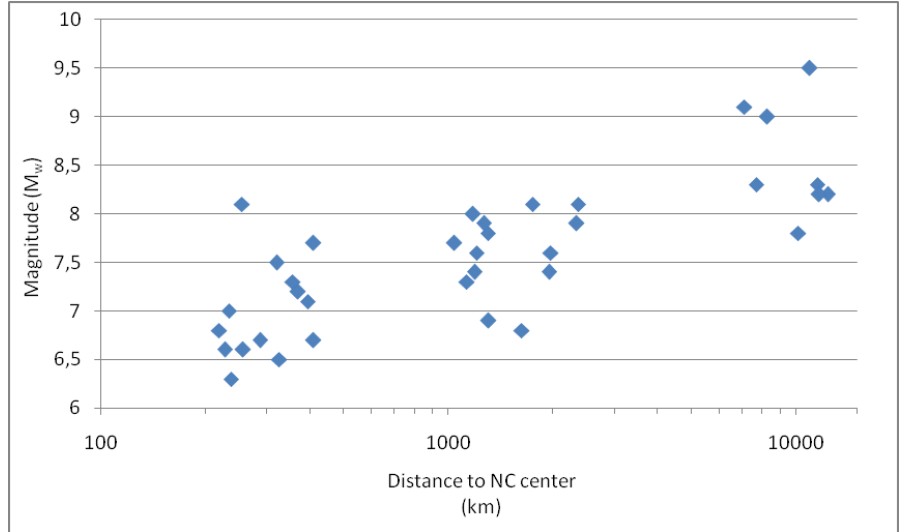


**Figure 13 : Graphic representation of the new catalogue showing earthquake's magnitude (Mw) as functions of distance between epicenter and New Caledonia.**




**Table 1 : Permanent tide gage locations and pressure sensors in New Caledonia.**

| Type | Longitude (°) | Latitude (°) | Name | in-operation date | First available high frequency (<5 min.) observation (SHOM) | Date of last available data |
|---|---|---|---|---|---|---|
| Tide gage stations | 166.436742 | -22.291478 | CHALEIX (Nouméa) | 24/02/1967 | None except pre 1967 paper records | 2005 |
| | 166.241528 | -21.613806 | THIO | 09/06/1967 | 23/04/2015 | |
| | 167.27869444 | -20.918472 | LIFOU | 08/09/1969 | 22/05/2013 | |
| | 166.561867 | -20.549816 | OUVEA | 06/04/1981 | 27/10/2011 | |
| | 166.68327646 | -21.982877 | OUINNE | 30/08/1981 | 17/05/2011 | |
| | 167.8771 | -21.5478 | MARE | 26/10/1982 | 22/04/2012 | |
| | 164.943122 | -20.691993 | HIENGHENE | 13/12/1983 | 23/02/2011 | |
| | 166.416218 | -22.241966 | NUMBO (Nouméa) | 29/07/2001 | 07/10/2010 | |
| Pressure sensors | 166.1832 | -22.285866 | Uitoe_04 | 24/06/2016 | 24/06/2016 | 14/11/2018 |
| | 166.48825 | -20.653333 | Ouvéa_02 | 23/09/2013 | 23/09/2013 | 04/04/2014 |
| | 165.322034 | -20.928805 | Poindimié_Tieti | 18/09/2013 | 01/11/2013 | 20/07/2016 |
| | 165.484807 | -20.891997 | Poindimié_Fourmi | 01/11/2013 | 01/11/2013 | 15/05/2017 |





**Table 2 : 25 tsunamis recorded in New Caledonia between September 30, 2009 and January 10, 2019. The 6 last events have not been reported as tsunamigenic in the NOAA NGDC catalogue.**

| | Earthquake time | Longitude (°) | Latitude (°) | Magnitude ($M_w$) | Hypocenter depth (km) | Location (subduction zones) | USGS ID | Distance to NC Center (km) | Observed maximum amplitude (cm) |
|---|---|---|---|---|---|---|---|---|---|
| NOAA NGDC/WDS Global Historical Tsunami Database | 2011-03-11T05:46:24.120Z | 142.373 | 38.297 | 9.1 | 29 | Japan | official20110311054624120_30 | 7089.41893 | 100 (Ouinné) |
| | 2011-07-06T19:03:18.260Z | 183.66 | -29.539 | 7.6 | 17 | Kermadec | usp000j48h | 1959.95312 | 20 (Ouinné) |
| | 2011-08-20T16:55:02.810Z | 168.143 | -18.365 | 7.2 | 32 | Vanuatu (South) | usp000j6r4 | 368.05544 | 25 (Ouinné) |
| | 2011-10-21T17:57:16.100Z | 183.762 | -28.993 | 7.4 | 33 | Kermadec | usp000j9nm | 1945.08139 | 10 (Ouinné) |
| | 2012-10-28T03:04:08.820Z | 227.899 | 52.788 | 7.8 | 14 | Cascadia | usp000juhz | 10142.895 | 5 (Hienghène) |
| | 2013-02-06T01:12:25.830Z | 165.114 | -10.799 | 8 | 24 | Vanuatu (North) - Santa Cruz earthquake (Sol.) | usc000f1s0 | 1171.90754 | 120 (Ouinné/Hienghène) |
| | 2014-04-01T23:46:47.260Z | 289.2309 | -19.6097 | 8.2 | 25 | Chile | usc000nzvd | 12326.6028 | 20 (Ouinné) |
| | 2014-04-12T20:14:39.300Z | 162.1481 | -11.2701 | 7.6 | 22.56 | Solomon | usc000phx5 | 1201.19814 | 10 (Ouinné/Hienghène) |
| | 2014-04-13T12:36:19.230Z | 162.0511 | -11.4633 | 7.4 | 39 | Solomon | usc000piqi | 1185.29192 | 15 (Ouinné/Hienghène) |
| | 2015-09-16T22:54:32.860Z | 288.3256 | -31.5729 | 8.3 | 22.44 | Chile | us20003k7a | 11497.7199 | 40 (Ouinné) |
| | 2016-12-08T17:38:46.280Z | 161.3273 | -10.6812 | 7.8 | 40 | Solomon | us20007z80 | 1296.63234 | 70 (Hienghène) |
| | 2016-12-09T19:10:06.840Z | 161.1316 | -10.749 | 6.9 | 19.73 | Solomon | us20007zlq | 1298.72096 | 10 (Hienghène) |
| | 2016-12-17T10:51:10.500Z | 153.5216 | -4.5049 | 7.9 | 94.54 | PNG | us200081v8 | 2328.06789 | 10 (Hienghène) |
| | 2017-09-08T04:49:19.180Z | 266.1007 | 15.0222 | 8.2 | 47.39 | Mexico | us2000ahv0 | 11598.7567 | 15 (Ouinné) |
| | 2017-10-31T00:42:08.720Z | 169.1485 | -21.6971 | 6.7 | 24 | Vanuatu (South) | us1000aytk | 286.21716 | 30 (Ouinné) |
| | 2017-11-01T02:23:57.670Z | 168.8585 | -21.6484 | 6.6 | 22 | Vanuatu (South) | us1000azjt | 255.733224 | 15 (Ouinné) |
| | 2017-11-19T22:43:29.250Z | 168.6715 | -21.3246 | 7 | 10 | Vanuatu (South) | us2000brlf | 232.901093 | 70 (Ouinné) |
| | 2018-08-29T03:51:56.100Z | 170.1262 | -22.0295 | 7.1 | 21.43 | Vanuatu (South) | us1000giaz | 392.401558 | 40 (Ouinné) |
| | 2018-12-05T04:18:08.410Z | 169.4181 | -21.9558 | 7.5 | 10 | Vanuatu (South) | us1000i2gt | 319.316922 | 200 (Isle of Pines) |
| | 2018-10-16T01:03:43.580Z | 169.5217 | -21.7427 | 6.5 | 17 | Vanuatu (South) | us1000hclz | 325.124924 | 10 (Ouinné) |
| Local tide gages only | 2011-05-10T08:55:08.930Z | 168.226 | -20.244 | 6.8 | 11 | Vanuatu (South) | usp000j1a8 | 218.199351 | 5 (Ouinné) |
| | 2012-03-09T07:09:50.950Z | 169.613 | -19.125 | 6.7 | 16 | Vanuatu (South) | usp000jfzj | 408.345795 | 10 (Ouinné) |
| | 2015-11-18T18:31:04.570Z | 158.4217 | -8.8994 | 6.8 | 12.59 | Solomon | us10003zcp | 1620.92574 | 20 (Ouinné) |
| | 2017-11-19T09:25:48.730Z | 168.6729 | -21.6377 | 6.3 | 14 | Vanuatu (South) | us2000brbk | 236.606262 | 10 (Ouinné) |
| | 2017-11-19T15:09:02.880Z | 168.5984 | -21.5027 | 6.6 | 13 | Vanuatu (South) | us2000brgk | 226.778385 | 15 (Ouinné) |





**Table 3 : List of tsunamis and associated seismic origins reported in New Caledonia from Soloviev and Go (1974) and Sahal et al. (2010). Observed maximum amplitudes have been estimated from witness observations.**

| Area concerned | Tsunami arrival in New Caledonia (Sahal et al., 2010) | Earthquake magnitude (Sahal et al., 2010) | Source of tsunami | Earthquake time (UTC) | Epicenter coordinates (USGS) Longitude (°) | Epicenter coordinates (USGS) Latitude (°) | USGS earthquake magnitude (Mw) | Hypocenter depth (km) | USGS ID | Distance to NC center (km) | Observed maximum amplitude (cm) |
|---|---|---|---|---|---|---|---|---|---|---|---|
| Local | 28/03/1875 | 8.1-8.2* | Vanuatu (South) | 1875-03-28T12:00:00.000 | | | | no information | | | > 250 (Lifou) |
| Local | 21/09/1920 | 8 | Vanuatu (South ) | 1920-09-20T14:39:03.000Z | 168,523 | -20,088 | 8.1 | 25 | iscgem912618 | 253,7432653 | 100-500 (Ouvéa) |
| Local | 17/05/1995 | 7.7 | E of Walpole Is. | 1995-05-16T20:12:44.220Z | 169,9 | -23,008 | 7.7 | 20,2 | usp0006xg1 | 408,0067327 | 50 (Maré) |
| Local | 28/12/2003 | 7.3 | Vanuatu (South) | 2003-12-27T16:00:59.450Z | 169,766 | -22,015 | 7.3 | 10 | usp000cg90 | 355,7768315 | 50 (Maré) |
| Regional | 04/10/1931 | 7 | Solomon | 1931-10-03T19:13:22.000Z | 161,11 | -11,117 | 7.9 | 15 | iscgem907054 | 1262,799956 | 150 (Hienghène) |
| Regional | 19/07/1934 | 7.8 | Vanuatu (North ) | 1934-07-18T19:40:19.000Z | 166,977 | -11,936 | 7.7 | 10 | iscgem905046 | 1038,560921 | 130 (Touho, Poindimié, Hienghène) |
| Regional | 21/07/1934 | 7 | Vanuatu (North ) | 1934-07-21T06:18:22.000Z | 165,89 | -11,129 | 7.3 | 15 | iscgem905065 | 1128,20392 | 100 (Touho) |
| Regional | 02/04/2007 | 8 | Solomon | 2007-04-01T20:39:58.710Z | 157,043 | -8,466 | 8.1 | 24 | usp000f83m | 1743,492778 | 200 (Hienghène) |
| Trans oceanic | 05/11/1952 | 9 | Kamchatka | 1952-11-04T16:58:30.000Z | 159,779 | 52,623 | 9 | 21,6 | official195211041658 30_30 | 8248,831736 | 200 (Yaté) |
| Trans oceanic | 23/05/1960 | 9,5 | Chile | 1960-05-22T19:11:20.000Z | 286,593 | -38,143 | 9,5 | 25 | official19600522191911 20_30 | 10943,44482 | 100 (Yaté) |
| Trans oceanic | 15/11/2006 | 8,3 | Kuril | 2006-11-15T11:14:13.570Z | 153,266 | 46,592 | 8,3 | 10 | usp000exfn | 7667,656657 | 50 (Ouinné) |
| Trans oceanic | 30/09/2009 | 8,1 | Tonga | 2009-09-29T17:48:10.990Z | 187,905 | -15,489 | 8,1 | 18 | usp000h1ys | 2355,426043 | 50 (Ouinné) |

* Information from Ioualalen and Pelletier (2017)

