# Peer review of "Update of the tsunami catalogue of New Caledonia using a decision table based on seismic data and maregraphic records."

_Natural Hazards and Earth System Sciences, 2019_

## Referee Comment (RC1) · Anonymous Referee #1 · 21 Mar 2019

Review of Roger et al 2019 manuscript (https://doi.org/10.5194/nhess-2019-36) The paper describes an interesting attempt to upgrade regional tsunami catalog based on some systematic approach. This approach is to formulate some quantitative criteria for the search of potential tsunamigenic events in the centennial USGS earthquake catalog that could affect the New Caledonia coast and then search for confirmation of tsunami observations on the New Caledonia coast in the available mareograph data and witness reports. So far, most of regional and national tsunami catalogs are updated on some other approach that starts from reported observation of unusual seas level oscillations at some point of the coast under consideration and/or the search local run-up or tide-gage data for already known tsunamigenic event affected other territories.

Application of this approach turned out to be rather productive and resulted in 25 new tsunami events that were recorded in New Caledonia during the last 10 years (from 2009 to 2019), that were added to 12 events previously known for this territory for the whole historical period (since 1875). Thus, this approach allowed more than double the total number of historical tsunamis observed in New Caledonia.

I think that the manuscript can be accepted as it is after some minor corrections (mostly technical) that are listed below.

Page 1 Line 34 Reference for Pelletier and Louat, 1989 is absent in the References List Page 1 Line 34 Reference for Bevis et 1995 is absent in the References List Page 1 Line 34 Reference for Pelletier et 35 al., 1998 is absent in the References List Page 1 Line 35 Reference for Calmant et al., 2003 is absent in the References List Page 2 Line 48 Reference for Soloviev and Go (1974) should be given as Soloviev and Go (1984) Page 3 Line 78 Reference for Rabinovich and Thomson, 2004 should be given as Rabinovich and Thomson, 2007 Page 4 Line 126 Reference for Aucan et al., 2017 is absent in the References List Page 4 Line 130 Reference for Aucan et al., 2017 is absent in the References List Page 4 Line 136 Reference for Aucan et al., 2018 is absent in the References List Page 8 Line 275 Reference for Herbitz et al. (2014) is absent in the References List Page 12. Figure 2 displays focal depth/magnitude chart for 744 tsunamigenic earthquakes around the World from 365AD to 2018. However, both parameters can be determined only for the instrumental period (that is since 1900 and with more or less reliable accuracy since 1964). For pre-instrumental period both parameters are usually the result of an expert judgment. I would suggest to re-plot this graph using only the events for the instrumental period.

Page 13 In Figure 4 meaning "P" is unclear. If it is the focal depth, it should be marked as "F'" as in the text on page 4 line 116. Page 21. In the Table 2 in the column "Distance to NC center" distance is given with an accuracy down to cm that is meaningless. It should be rounded to km. Page 22. In the Table 3 in the column "Distance to NC center" distance is given with an accuracy down to mm that is

meaningless. It should be rounded to km.

Please also note the supplement to this comment:
https://www.nat-hazards-earth-syst-sci-discuss.net/nhess-2019-36/nhess-2019-36-RC1-supplement.pdf

---

## Author Comment (AC1) · 27 Mar 2019

Dear Referee,

We are pleased to provide answers to your comments below:

You detailed about references lacking to our manuscript references list. It is a mistake from us to have removed those references in our review process in Word software instead of keeping them. It is now fixed.

Concerning Figure 2, we fully agree with you; it was a mistake from us in the text and the beginning time of the considered events has been changed to "January 1,

1970"; the data used for preparing the graph have been also changed to keep only the instrumental period, thus only 440 events are now considered.

Concerning Figure 4, "P" has been changed by "F" to be in agreement with the manuscript.

Table 2 and Table 3 column "Distance to NC center" has been corrected: it is true that keeping decimals is a nonsense for such distances in hundreds of kilometers.

Thank you for providing comments and your detailed review.

Best regards,

The authors.
* * *
[Figure]

**Fig. 1.** Relationship between earthquake focal depth and moment magnitude for 440 tsunamigenic events of the NGDC/WDS Global Historical Tsunami Database.

**Fig. 2.** Decision algorithm to select events automatically.

---

## Author Comment (AC2) · 27 Mar 2019

Dear Referee,

We are pleased to provide answers to your comments below:

You detailed about references lacking to our manuscript references list. It is a mistake from us to have removed those references in our review process in Word software instead of keeping them. It is now fixed.

Concerning Figure 2, we fully agree with you; it was a mistake from us in the text and the beginning time of the considered events has been changed to "January 1, 1970";

[Figure]

the data used for preparing the graph have been also changed to the instrumental period (since january 1, 1970): only 440 events are kept.

Concerning Figure 4, "P" has been changed by "F" to be in agreement with the manuscript.

Table 2 and Table 3 column "Distance to NC center" has been corrected: it is true that keeping decimals is a nonsense for such distances in hundreds of kilometers.

Thank you for providing comments and your detailed review.

Best regards,

The authors.
* * *
[Figure]

**Tsunamigenic events: moment magnitude vs. focal depth**

**Fig. 1.** Relationship between earthquake focal depth and moment magnitude for 440 tsunamigenic events of the NGDC/WDS Global Historical Tsunami Database.

**Fig. 2.** Decision algorithm to select events automatically.

---

## Referee Comment (RC2) · Anonymous Referee #2 · 15 Apr 2019

This paper is an important contribution to the building of tsunami databases in areas rarely impacted by catastrophic events, but which lay close to significantly active subduction zones. The update proposed in the study is worth being published, and also shared among the scientists and general public to increase awareness.

From a scientific point of view some aspects of the paper have to be improved or clarified, some sections should be better hierarchized, thus it may be published after a minor revision is done.

The title indicates a decision table, while the text relies on an algorithm. However, the title seems more appropriate since the procedure is a conditional extraction of a

database.

The abstract should be more precise on the dataset obtained, and should be representative of the whole content. The 1960 tsunami evidenced has to be mentioned, as well as near l.52.

In the beginning of section 2.4, it should be added that the tsunami amplitude also relies on the rupture characteristics and dynamics, not only on the geometry. Indeed tsunami earthquakes could also be foreseen (are they any documented in the regional sources?).

In section 2.4, the choice of the distance criterion should be more discussed. Does the average strike of a given subduction play a role also? If no, it should also be stated.

In section 2.5, the search procedure seems to be applied to the whole dataset, or was there any succession of conditional tests (first magnitude and depth, then distance)? In the latter case the four boxes should not be shown at the same level in the figure 2, but after each other. Later on the 6-digit accuracy after the comma does not make sense to describe the box for the barycenter estimation.

All the locations described in section 2.6 are not reported in Figure 5: Grande Terre, Loyalty, Noumea (even though Chaleix and Numbo are displayed) are missing (and subsequently referred locations such as Isle of Pines, Ponerihouen, Canala, or Mou in Lifou, as well). Here a first mention is made of the arrival azimuth (see remark above on the distance criterion), has it finally to be considered? Since some tsunamis can also be important on the lee side of islands, the azimuth is probably not the only reason, and coastal and reef conditions should be mentioned. And in addition the 1960 tsunami was finally well observed in Noumea. The paragraph should be improved to be more consistent.

Regarding the arrival azimuths from NZ or Papua New Guinea, the reader cannot easily figure out where these azimuths are on the figure. At least some arrows could be

added.

In the beginning of section 3, it is now stated that magnitudes above 6.3 are considered. This is not consistent with the work previously presented with the criterion above 6: what was the use to consider 6.0 in the previous section? The section 2.3 could be modified to add the percentage corresponding to Mw < 6.3. Overall the reasoning does not seem logical. Keeping from the beginning magnitude above 6.3 could have been sufficient.

The following presents the most important results, but the reading is not always easy. The section 3.1 is a mix of 1) cross checking catalogues and 2) checking tide gauge data, and finally 3) checking tide gauge data independently from the catalogues (which could actually be even the first step in the reasoning), but overall the text should be better hierarchized. Also to help the reading, two bullets could be added on lines 159 and then 164, to identify the two periods that are investigated for the 32 events remaining.

The end of section 3.1 establishes percentages for the different periods studied, but the conclusion drawn is not straightforward: what does the factor 10 increase mean? That the Sahal catalogue was not complete (but based on a different approach)? Could it be commented further?

Then the new data are presented and this a very important part, showing how tide gauge data are essential to better understand tsunami impact. Ouinne is identified as an amplifying place during the 2015 tsunami from Chile. Is the relative amplification in Chaleix compared to Numbo due to different locations from the open sea? Their detailed locations are not described.

The recent Dec. 2018 event is important also to raise awareness. This is not the core purpose of the paper, but it could be worth mentioning the min and max horizontal inundation and drawdown observed during the tsunami.

Finally the section 3.3 comes back to local observations independently from catalogues. If the tide gauge data are available for 2010 in Numbo, it could be shown, a 10 min sampling could be sufficient. In that case the use of a spectrogram could be helpful (as well for all other data, in order to also quantify the bay amplifications). The 1960 is very interesting; however the following sheet is not shown (after 16:00), thus the detiding is not very accurate and the remaining trend is probably affected by a boundary effect. By the way the theoretical tide used should be explained (or is this a filter?). If further temporal analysis is not possible, it should at least be mentioned. It would be very informative to have the whole sequence that probably lasted more than 24 h.

In the section 4, the overall Pacific setting is presented, but actually the figure 12 could even be put at the beginning of the paper when the catalogues are presented. The 2016 Papua New Guinea is reported with a depth of 100 km and a small tsunami was triggered. This poses the question of the figure where depths below 100 km are kept while the method should have rejected the corresponding event. Note that GMT Harvard put it at a 50 km depth.

The figure 13 could have been completed with a graph of tsunami heights as a function of the magnitude that can be useful for pre operational procedures. Again, the distance criterion is not the most relevant to analyze the dataset, since the orientation of the main energy spread is at least as decisive to produce a tsunami. In addition distant earthquakes with magnitude lower than 7.7 (possibly from 7.4 to 7.6) and slow ruptures (tsunami earthquakes) could also pose some risk.

The paper does not mention any paleotsunami research although the area would probably deserve some investigations, this could be mentioned.

* Some other remarks: l.36-37: the maximum magnitude of 8.2 is for instrumental events only, and this is already for strong earthquakes. The "although" does not sound accurate. On the contrary, it could be stated that magnitudes well above 8.2 to 8.5 are

possible in the area and most probably tsunamigenic.

l.55: the term "algorithm" could also be named a conditional test, as it is mostly the case in this paper

l.69: the box described here is not easy to figure out without a map, it may be also commented as describing the whole Pacific extension? And a 3 digit accuracy after coma is not necessary in the box.

l.98: the tsunami wavelength is essentially related to the fault width, and to the rupture dynamics. This should be added.

l.222: "more so than at" sentence to be revised?

l.232: coma to be removed before foreshocks

l.282: isle of Pins was isle of Pines elsewhere, it should be unified (and added on Figure 1)

l.325: it is mentioned that less people used to live close to the shores but the first part of the sentence refers to Sept 2009 as a turning point: has the population density that changed within 10 years? Or is it for older periods?

* Figures:

Figure 1: the label "inactive subduction zone" does not seem to be associated with any structure of the figure, so is it useful? If it is the case, it should be made clearer (maybe it is hidden beneath some earthquakes?). Also the legend could also describe the color scale. Convergence rates could also be added.

Figure 2: two lines Mw = 6 (or 6.3?) and depth = 100 could be highlighted to delimit the dataset finally used (or rejected). And the legend could describe the procedure applied as a filter. Otherwise the figure itself does not provide any message.

Figure 4: see the comment above.

Figure 5: the names of the location should be enlarged, and at least two locations described in the text are missing: Ile des Pins, as well as Yate (see also comments above).
* * *

---

## Author Response (AR1)

Dear Editor,

We are pleased to provide the final draft of our manuscript entitled "**Update of the tsunami catalogue of New Caledonia using a decision table based on seismic data and maregraphic records**". Answers to the referees' comments are listed below.

**Referee #1**

The referee detailed about references lacking to our manuscript references list. It is a mistake from us to have removed those references in our review process in Word software instead of keeping them. It is now fixed.

Concerning Figure 2, we fully agree with him; it was a mistake from us in the text and the beginning time of the considered events has been changed to "January 1, 1970"; the data used for preparing the graph have been also checked.

Concerning Figure 4, "P" has been changed by "F" to be in agreement with the manuscript.

Table 2 and Table 3 column "Distance to NC center" has been corrected: it is true that keeping decimals is a nonsense for such distances in hundreds of kilometers.

**Referee #2**

The referee mentioned the fact that decision table is more appropriate than algorithm and he is right. We agree and change the word in the manuscript.

Considering the 1960 tsunami, we mentioned it in the abstract as well as at the end of the part 1.

Comment about role of the rupture characteristics and dynamics on the tsunami generation has been considered in section 2.4.

As requested, the choice of the distance criterion linked to the magnitude criterion has been more discussed in section 2.4. We are aware of the role played by the strike at a given subduction zone, but the objective was to select all the events able to trigger a tsunami being recorded on tide gauges in New Caledonia.

In section 2.5 the search procedure is applied to the whole dataset. There is no succession of conditional tests. Of course, we have removed the 6-digit accuracy of the columns in table 2 and 3, as requested by another reviewer.

The locations of Grande Terre, Loyalty Islands, Nouméa, Ponérihouen, Isle of Pines, Canala and Mou have been added to figure 5. Concerning the potential of tsunami record in Nouméa (along the West coast), paragraph 2.6 has been improved as requested.

We cut off the sentence about the arrival azimuths of tsunamis in section 2.6.

We fully agree with the referee about the confusion on the magnitude criterion of 6 or 6.3 at the beginning of section 3. It doesn't make any change on the results since the events between 6 and 6.3 were located outside of the Pacific Ocean and/or they did not produce a sufficient tsunami to be recorded in New Caledonia, we decided to look only at earthquakes of magnitude Mw > 6.3 as he indicated.

Section 3.1 has been re-organized and better hierarchized as the referee recommended.

The sentence dealing with the different percentages for the 2 studied periods (1900-->2009, 2009--> 2019) has been improved. By the way, the Sahal catalogue is based on a different approach and our methodology doesn't help to improve the catalogue before 2009.

Although we noticed a difference of signal amplitude 3 times more important at Chaleix than at Numbo during one event, we have not explored the issue further as the two tide gauges are in different bays and are likely affected by different resonance mechanisms.

The data concerning the December 5, 2018 tsunami are kept for a further publication focusing on this event.

We added a sentence in the sea-level data section to describe our method for de-tiding. For the Nouméa tide gauge, it is done with a multiyear time-series and it is therefore very accurate.

Figure 12 only shows the results of our study: the location of the earthquakes having triggered a tsunami recorded in New Caledonia. By analyzing the Pacific data about tsunamis (NOAA NGDC catalogue) we found that no tsunami triggered by distant earthquake located far from Nouméa (> 2500 km) and deeper than 100 km was recorded in New Caledonia. The same way, no regional tsunami has been related to an earthquake deeper than 100 km and with magnitude lower than 7.5. The scale bar has been modified in accordance (depth > 100 km has been removed).

Unfortunately, the scarce tsunami amplitude data along the coast of New Caledonia does not allow yet to produce a relevant graph of tsunami heights as a function of the earthquake magnitude to be useful for pre-operational procedures. We still think that the distance criterion is a relevant to analyze the dataset. Note that a shallow (< 100 km) earthquake with magnitude higher than 7.5 is kept in our process.

Concerning paleotsunami research, it is true that we do not mention that point because there is no published research on this topic in New Caledonia. A paleotsunami research project has just been funded and will be held in a few month in 2019.

Answers to the other remarks:

- L.36-37: the sentence has been modified as requested.

- L. 55: corrected.

- L. 69: corrected.

- L. 98: modified.

- L. 222: corrected.

- L. 232: corrected.

- L. 282: corrected.

- L. 325: modified.

- Figure 1: The inactive subduction zones are indicated on the figure as red dashed lines. Convergences rates have been added. Legend has been modified.

- Figure 2: improved with the two requested lines.

- Figure 4: done.

- Figure 5: added.

Best regards,

The authors.

[revised manuscript text omitted]

---

## Author Response (AR2)

Dear Editor,

We are pleased to provide the final draft of our manuscript entitled "**Update of the tsunami catalogue of New Caledonia using a decision table based on seismic data and maregraphic records**" having considered you comments about the reference list and the tables:

* Digits

The number of digits for the longitude, latitude and focal depths in the 3 tables have been unified as requested.

* Reference list

Each reference of the reference list has been checked individually:

- some mistakes have been corrected

 - Baillard et al. (2015) has been removed from the list

 - In the text, "Bevis et 1995" has been corrected --> "Bevis et al., 1995"

 - Table 3: "Ioualalen and Pelletier (2017)" has been corrected --> "Ioualalen et al. (2017)"

 - the reference of Soloviev and Go (1974) has been improved

- the format has been unified in agreement with NHESS specifications

Best regards,

The authors.

---

## Author Response (AR3)

Dear Editor,

We are pleased to provide the final draft of our manuscript entitled "**Update of the tsunami catalogue of New Caledonia using a decision table based on seismic data and maregraphic records**" having considered you comment about the tables: the commas have been replaced by points as requested.

Best regards,

The authors.